# Toxicity Reduction of *Euphorbia kansui* Stir-Fried with Vinegar Based on Conversion of 3-*O*-(2′*E*,4′*Z*-Decadi-enoyl)-20-*O*-acetylingenol

**DOI:** 10.3390/molecules24203806

**Published:** 2019-10-22

**Authors:** Qiao Zhang, Yi Zhang, Shi-Kang Zhou, Kan Wang, Min Zhang, Pei-Dong Chen, Wei-Feng Yao, Yu-Ping Tang, Jian-Hua Wu, Li Zhang

**Affiliations:** 1Jiangsu Key Laboratory for High Technology Research of TCM Formulae, National and Local Collaborative Engineering Center of Chinese Medicinal Resources Industrialization and Formulae Innovative Medicine and Jiangsu Collaborative Innovation Center of Chinese Medicinal Resources Industrialization, Nanjing University of Chinese Medicine, Nanjing 210023, China; zhangqiao@njucm.edu.cn (Q.Z.); zhangyi@njucm.edu.cn (Y.Z.); zsk6428249@163.com (S.-K.Z.); wangkan86@hotmail.com (K.W.); nzyzhm@163.com (M.Z.); chenpeidong1970@163.com (P.-D.C.); yaowf@njucm.edu.cn (W.-F.Y.); 2Key Laboratory of Shaanxi Administration of Traditional Chinese Medicine for TCM Compatibility, Shaanxi University of Chinese Medicine, Xi’an 712046, China; yupingtang@sntcm.edu.cn (Y.-P.T.); wu0700@126.com (J.-H.W.)

**Keywords:** *Euphorbia kansui*, diterpenoids, vinegar processing, zebrafish toxicity

## Abstract

The dried roots of *Euphorbia kansui* S.L.Liou ex S.B.Ho have long been used to treat edema in China. However, the severe toxicity caused by *Euphorbia kansui* (EK) has seriously restricted its clinical application. Although EK was processed with vinegar to reduce its toxicity, the detailed mechanisms of attenuation in toxicity of EK stir-fried with vinegar (VEK) have not been well delineated. Diterpenoids are the main toxic ingredients of EK, and changes in these after processing may be the underlying mechanism of toxicity attenuation of VEK. 3-*O*-(2′*E*,4′*Z*-decadienoyl)-20-*O*-acetylingenol (3-O-EZ) is one of the diterpenoids derived from EK, and the content of 3-O-EZ was significantly reduced after processing. This study aims to explore the underlying mechanisms of toxicity reduction of VEK based on the change of 3-O-EZ after processing with vinegar. Based on the chemical structure of 3-O-EZ and the method of processing with vinegar, simulation experiments were carried out to confirm the presence of the product both in EK and VEK and to enrich the product. Then, the difference of peak area of 3-O-EZ and its hydrolysate in EK and VEK were detected by ultra-high-performance liquid chromatography (UPLC). Furthermore, the toxicity effect of 3-O-EZ and its hydrolysate, as well as the underlying mechanism, on zebrafish embryos were investigated. The findings showed that the diterpenoids (3-O-EZ) in EK can convert into less toxic ingenol in VEK after processing with vinegar; meanwhile, the content of ingenol in VEK was higher than that of EK. More interestingly, the ingenol exhibited less toxicity (acute toxicity, developmental toxicity and organic toxicity) than that of 3-O-EZ, and 3-O-EZ could increase malondialdehyde (MDA) content and reduce glutathione (GSH) content; cause embryo oxidative damage by inhibition of the succinate dehydrogenase (SDH) and superoxide dismutase (SOD) activity; and induce inflammation and apoptosis by elevation of IL-2 and IL-8 contents and activation of the caspase-3 and caspase-9 activity. Thus, this study contributes to our understanding of the mechanism of attenuation in toxicity of VEK, and provides the possibility of safe and rational use of EK in clinics.

## 1. Introduction

Processing of traditional Chinese medicines (TCM) is a traditional technique of “PaoZhi (Chinese: 炮制)”, which has been applied in China for thousands of years. The processing of TCM has an important influence on clinical safe applications of toxic TCM, as different changes can be effected after processing, such as reduced toxicity and improved efficacy. Chemical compositions of some drugs can change after processing, which leads to changes in efficacy and toxicity. For example, with *Aconitum* tubers and *Strychnos nux-vomica* L., studies have shown that the toxicity of *Aconitum* tubers was reduced after processing, which was due to that the diester-diterpenoid alkaloids (aconitine and hypaconitine) being hydrolyzed into monoesters-diterpenoid and amines-diterpenoid alkaloids [1,2], and the toxicity of *Strychnos nux-vomica* L. was reduced after processing, which was caused by the indole alkaloids (strychnine and brucine) converted to less toxic nitrogen oxides [3,4]. Meanwhile, the contents of aconitine, hypaconitine, strychnine and brucine all decreased after processing. Therefore, confirming the compositional changes of the drug and comparing the differences in efficacy and toxicity before and after being processed could provide a basis for revealing the processing mechanism of TCM.

Kansui (Chinese: 甘遂), the dried roots of *Euphorbia kansui* S.L.Liou ex S.B.Ho (EK), recorded in Shennong-Bencao, is widely used in TCM for clinical treatment of edema, ascites, cancer, HIV, intestinal obstruction and asthma [5,6,7]. However, the severe hepatotoxicity and gastrointestinal toxicity caused by EK has restricted its clinical applications [8]. Previous studies showed that the structure type of the compounds in EK including diterpenes, triterpenes, flavonoids, phenolic and acids. Among them, diterpenoids (ingenane and jatrophane type) and triterpenoids are the major bioactive components from EK, and are responsible for the toxicity effect of EK [9,10]. In contrast to the triterpenoids, the diterpenoids of EK exhibit stronger cytotoxic activity, indicating that EK may exert systemic toxicity effect mainly though the actions of diterpenoids. Notably, 3-*O*-(2′*E*,4′*Z*-decadienoyl)-20-*O*-acetylingenol (3-O-EZ), a major diterpenoid of EK, plays a significant role in the process of toxicity effect of EK [11,12].

Recent studies have provided compelling clues that EK was processed with vinegar before being used in the clinic to minimize the toxicity of EK [5,8]. As we know, the chemical compositions determine the efficacy and toxicity of TCM to some extent, so the reduced toxicity of EK after processing is bound to cause changes in composition. Research has shown that the LC_50_ value of ethyl acetate extracted from VEK (6.62 ± 1.24 μg/mL) was greater than EK (2.78 ± 0.86 μg/mL) in embryos of zebrafish, and the content of 3-O-EZ decreased by 12.79% in ethyl acetate of VEK compared with EK, suggesting that the change of diterpenoids is correlated with the attenuated toxicity of VEK [7]. Moreover, numerous studies have demonstrated that the content of diterpenoids in EK was mostly reduced after being processed with vinegar [13,14]. Nevertheless, the detail mechanisms of attenuation in toxicity of VEK mediated by processing with vinegar have not been sufficiently explored as yet.

At present, it is generally known that the zebrafish is a valuable biological model for toxicity research [15,16]. Compared to other animal experimental models, the zebrafish model is much less time consuming and comes with a lower dosage [7]. In particular, zebrafish embryos have been well recognized as an alternative to traditional experimental animals [17].

Based on the above reports, it would be postulated that the ester bond of diterpenoids in EK may be broken and converted into less toxic compounds after processing, and changes in structure and content of diterpenoids may contribute to the toxicity reduction of VEK. Thus, in the present study, the conversion reaction of 3-O-EZ was conducted by simulating vinegar processing of EK for validating the changes in structure diterpenoids. Meanwhile, determination of the peak areas of 3-O-EZ and its hydrolysate in EK and VEK was conducted. Finally, acute toxicity, developmental toxicity and organic toxicity of 3-O-EZ and its hydrolysate on zebrafish embryos were compared involved in oxidative damage, immune response and cell apoptosis. This study could provide evidence for the material basis and mechanism of attenuation in toxicity of VEK.

## 2. Results

### 2.1. Identification of Hydrolysate

The positive and negative ion mass spectrum of hydrolysate are shown in Appendix A. The positive ion mass spectrometry cracking peak of hydrolysate was provided as follows: *m*/*z* 371 [M + Na]^+^, 349 [M + H]^+^, 331 [M + H − H_2_O]^+^, 313 [M + H − 2H_2_O]^+^, 295 [M + H − 3H_2_O]^+^, 267 [M + H − 3H_2_O − CO]^+^. The negative ion mass spectrometry cracking peak of hydrolysate was provided as follows: *m*/*z* 347 [M − H]^−^, 329 [M − H − H_2_O]^−^, 301 [M − H − H_2_O − CO]^−^. The possible process of cracking is shown in Appendix A. The ^1^H-NMR and ^13^C-NMR data of hydrolysate are shown in Appendix A. Compared to the literature [18,19], the hydrolysate was determined to be ingenol and the conversion process between compounds 3-O-EZ and ingenol is shown in Figure 1.

### 2.2. Peak Areas Comparison of 3-O-EZ and Ingenol in EK and VEK

As shown in Figure 2, the peak area of the 3-O-EZ in EK was 113577 at 254 nm, which was higher than that of VEK (81855), whereas the peak area of the ingenol in VEK was 33962 at 210 nm, which was greater than that of EK (12835).

### 2.3. Comparison of Toxicity of 3-O-EZ and Ingenol

#### 2.3.1. Acute Toxicity

The acute toxicity curves of 3-O-EZ are shown in Appendix A. 3-O-EZ presented definite toxicity with LC_100_ value of 0.990 μg/mL and LC_0_ value of 0.166 μg/mL, and the LC_50_ value of 3-O-EZ was 0.412 ± 0.01 μg/mL. The pre-experimental result showed that the LC_100_ and LC_0_ of the ingenol group on zebrafish embryos was greater than 100 μg/mL when compared with those in 3-O-EZ group, suggesting that ingenol was less toxic to embryos than 3-O-EZ. Thus, the LC_50_ of ingenol was no longer performed.

#### 2.3.2. Developmental Toxicity

Compared to the control group, 3-O-EZ group can reduce the heart rate of zebrafish embryos with dose-dependent relationship, while the ingenol group was not changed (Table 1). As shown in Figure 3, compared to the control group, three dose groups of 3-O-EZ appeared pericardial edema, and the high and low dose groups of 3-O-EZ exhibited scoliosis. However, all ingenol treatment groups showed no obvious pericardial edema and scoliosis compared with the control group.

#### 2.3.3. Organic Toxicity

The distance between the Sinus venosus (SV) and Bulbus arteriosus (BA) of the heart was determined (Table 1). Compared to the control group, 3-O-EZ group can increase the distance of SV-BA with dose-dependent changes, while the changes of three ingenol groups were not observed. Acridine orange (AO) staining was used to estimate whether 3-O-EZ and ingenol groups can cause apoptosis in zebrafish embryos (Figure 4). Compared to the control group, increased fluorescence intensity and areas were observed in the embryos under 3-O-EZ treatment groups with dose-dependence, while no significant change was observed in all ingenol treatment groups.

The gastrointestinal motility rates of zebrafish embryos in different groups were determined (Table 2). Compared to the control group, 3-O-EZ groups can retard the gastrointestinal motility rate with dose-dependence, while no significant change was observed in all ingenol treatment groups. The relative area of zebrafish gastrointestinal (GI) in each group was measured. As shown in Figure 5, compared to the control group (defined as 100%), the mean relative GI area of zebrafish treated with 3-O-EZ at concentrations of high, moderate and low were 62%, 87% and 95%, respectively (Table 2), while no significant change was observed in all ingenol treatment groups (104%, 104% and 103%, respectively).

#### 2.3.4. Measurement of SDH, SOD Activities, GSH and MDA Content

Compared to the control group, the activities of succinate dehydrogenase (SDH), superoxide dismutase (SOD), and glutathione (GSH) content were significantly decreased in the high (0.412 μg/mL) and moderate (0.206 μg/mL) 3-O-EZ groups. The content of malondialdehyde (MDA) was obviously increased in the high (0.412 μg/mL) and moderate (0.206 μg/mL) 3-O-EZ groups compared with the control group. Compared to the control group, the activities of SDH and SOD, GSH and MDA content almost exhibited no significant changes in the different ingenol groups, although the MDA content was increased in the moderate (0.206 μg/mL) ingenol group (Figure 6).

#### 2.3.5. Measurement of Caspase Activity, IL-2 and IL-8 Content

The contents of Interleukin-2 (IL-2) and Interleukin-8 (IL-8) were obviously higher in all 3-O-EZ treatment groups than in the control group, while all ingenol treatment groups showed no significant changes compared with the control group. The activities of caspase-3 and caspase-9 were increased in 3-O-EZ treatment groups than the control group, except for the low 3-O-EZ group, while there was no significant difference between the ingenol group and the control group in the effect of caspase-3 activity. The activity of caspase-9 was increased in the high and moderate ingenol groups compared with the control group (Figure 7).

#### 2.3.6. Gene Expression

The expression levels of sdha and GPx were significantly decreased at 0.412 and 0.206 μg/mL of 3-O-EZ, respectively, compared with the control group, while the ingenol groups showed no significant changes, except in the high (0.412 μg/mL) concentration ingenol group (Figure 8). The expression levels of TNF-α and IL-8 were significantly increased at 0.412 and 0.206 μg/mL of 3-O-EZ treatment, respectively, while no obvious enhancement of TNF-α and IL-8 was observed in the ingenol groups. The expression levels of the apoptosis-related genes were significantly up-related at 3-O-EZ group (0.412 μg/mL) compared with the control group, while there was no significant difference between the ingenol group and the control group in the levels of the apoptosis-related genes (Figure 8).

## 3. Discussion

Numerous studies confirmed that the contents of toxic components decreased after EK was processed with vinegar. However, what happened during the conversion of these components from VEK and the relationship between structure and toxicity were unknown. As EK had severe hepatotoxicity, gastrointestinal toxicity, irritation, etc., it needed to be processed in clinical applications to reduce its toxicity. According to the 2015 edition of the Chinese Pharmacopoeia and clinical practice, EK’s processing method is to be stir-fried with vinegar. Rice vinegar is a kind of Chinese medicine excipient and mainly contains acetic acid, lactic acid, tyrosine, etc. [5,20]. Previous studies had manifested that the content of 3-O-EZ was reduced in VEK, so, we speculated that 3-O-EZ in EK may convert into ingenol after processing. Thus, the conversion reaction of 3-O-EZ was conducted for validating the change of 3-O-EZ and enriching ingenol. The present UPLC result showed that both 3-O-EZ and ingenol were found in EK and VEK; however, the content of 3-O-EZ was reduced in VEK, and the content of ingenol was increased in VEK, indicating that the 3-O-EZ in EK could change into ingenol after processing with vinegar. The simulation reaction of 3-O-EZ demonstrated that the conversion process from 3-O-EZ to ingenol may be due to the fracture of the ester bond; meanwhile, the ingenol was obtained and enriched for toxicity evaluation on zebrafish embryos.

Zebrafish embryos are small and relatively transparent, and multiple organs can be observed directly under the microscope. The heart is an important organ in the development of zebrafish embryos [21], and the gastrointestinal tract plays a major role in mucosal homeostasis, barrier function and immunity [22]. However, abnormal heart rate, pericardial edema, and gastrointestinal motility lead to abnormality of embryonic development or even death [23,24]. In the present study, the LC_50_ value of ingenol (>100 μg/mL) was far greater than 3-O-EZ (0.412 ± 0.01 μg/mL), and zebrafish embryos treated with 3-O-EZ showed bradycardia and pericardial edema, increased SV-BA value, and decreased GI motility rate and mean relative GI area. While the above changes are not found in all ingenol treatments, these results indicated that 3-O-EZ possessed cardiotoxicity and gastrointestinal toxicity in zebrafish embryos, which may lead to the abnormal development of zebrafish embryos.

The activity of SDH binding to mitochondrial inner membrane represents the function of mitochondria [23]. Conversely, mitochondrial dysfunction and decreased SDH activity can lead to oxidative damage. Previous studies confirmed that the oxidative damage was considered as an important mechanism of organ damage [8], and the activity and expression levels of SOD and GSH content were related to the antioxidant capacity of cells [25,26]. Besides, MDA was a lipid peroxidation product produced by free radical attack, and the level of change in MDA reflected the degree of damage to cells [8,23]. In this experiment, the activity of SDH and related gene expression (sdha) were significantly decreased at 0.412 and 0.206 μg/mL of 3-O-EZ, respectively, that 3-O-EZ may lead to damage of mitochondrial function. In addition, the activity and expression levels of the antioxidant enzyme were significantly decreased at 0.412 and 0.206 μg/mL of 3-O-EZ, respectively, which may lead to an abnormality during embryonic development.

Furthermore, there was a close relationship between inflammation and oxidative damage. IL-2 is a cytokine with an autocrine survival and proliferation signal for T cells. IL-8 is an important chemotactic and pro-inflammatory cytokine, which has been confirmed to participate in immune regulation and play an important role in the regulation of inflammation [8,27,28]. In this study, the levels of IL-2, IL-8 and immune-related gene expression were markedly up-regulated in zebrafish embryos treatment with 3-O-EZ, suggesting that the EK induced inflammatory response was mediated by 3-O-EZ.

Previous studies demonstrated that oxidative damage and inflammation can lead to apoptosis [23]. The inhibition of anti-apoptotic protein Bcl-2 and activation of caspase-3, caspase-9 and Bax can induce apoptosis of cells, and the ratio of Bcl-2/Bax was a crucial element of apoptotic cell death [8]. In this study, the activity of caspase-3 and caspase-9 as well as immune-related gene expression were markedly up-regulated in zebrafish embryos treatment with 0.412 μg/mL 3-O-EZ. These results indicated that cell apoptosis was induced by 3-O-EZ in zebrafish embryos, and this may be the explanation for EK-induced abnormal embryonic development and even death.

Taken together, 3-O-EZ could inhibit the SDH and SOD activity, reduce the GSH content, and increase MDA content, and ultimately lead to embryo oxidative damage. Besides, 3-O-EZ can induce inflammation and apoptosis by increasing IL-2 and IL-8 contents and activating caspase-3 and caspase-9 activity. Moreover, the ingenol showed no obvious toxic effect and changes of zebrafish embryonic development. The 3-O-EZ exhibited toxic effect with a dose-dependent relationship on zebrafish embryos, and the toxicity of ingenol was less than 3-O-EZ. These results may be the reason for zebrafish embryo development abnormality or even death caused by 3-O-EZ.

All results mentioned above validated our previous assumption in the Introduction and may explain why vinegar-processing can significantly reduce toxicity of EK. However, we only evaluated the toxicity and did not evaluate the efficacy, and the structure-efficacy of 3-O-EZ and ingenol will be further studied in an H22 mouse hepatoma ascites model.

## 4. Materials and Methods

### 4.1. Chemicals and Reagents

3-*O*-(*2′E,4′Z*-Decadienoyl)-20-*O*-acetylingenol (3-O-EZ) was extracted from EK and purified in our laboratory (purity determined by HPLC-UV were more than 98%) [9]. Acetonitrile and formic acid of HPLC grade were purchased from Merck Co. (Darmstadt, Germany). Acetic acid, tetrahydrofuran and dichloromethane were of analytical grade. Ultra-pure water was purified by a Milli-Q academic water purification system (Millipore, Billerica, MA, USA). AO was purchased from Shanghai Macklin Biochemical Co. Ltd. (Shanghai, China). Calcein was purchased from Aladdin industrial Corporation (Shanghai, China).

### 4.2. Zebrafish Embryos

Zebrafish were purchased from Nanjing YSY Biotech Company LTD (Nanjing, China). As the previous research described [7], the embryos were collected for further analysis.

### 4.3. Materials

The dried root of EK was collected from Red River valley of Baoji, Shaanxi province of China, in October 2017 and was identified by Prof. Qinan Wu (Nanjing University of Chinese Medicine, Nanjing, 210023, China). The voucher specimen (20171015) has been deposited in the Herbarium of college of pharmacy, Nanjing University of Chinese Medicine (Nanjing, Jiangsu, China). A total of 100 g of cleaned EK was immersed in 30 g vinegar and stir-fried at 260 °C until slight scorched spots appeared to obtain VEK; then the dried EK and VEK were crushed into powder (65-mesh) [7].

### 4.4. Preparation and Analysis of Hydrolysate

#### 4.4.1. Conversion Reaction

3-O-EZ was accurately weighed (24.5 mg) and put into a 10 mL ampoule. Then, the appropriate amount of tetrahydrofuran was added to promote dissolving. Lastly, 2 mL of 3 mol/L acetic acid solution was added and stored in electro-thermostatic blast oven (EBO) (DHG-9023A, Shanghai, China) at 120 °C for 3 h. After the sample was cooled, it was extracted three times with dichloromethane and concentrated up to dryness to obtain hydrolysate.

#### 4.4.2. Purification and Analysis of Hydrolysate

The hydrolysate was purified by a Waters 1525 with a 2996 Diode Array Detector (DAD) detector (Waters, Milford, CT, USA). The structure of the hydrolysate was analyzed by ultra-high-performance liquid chromatography coupled to quadrupole time-of-flight mass spectrometry (UPLC-QTOF/MS), the UPLC-MS analysis was performed on a Waters ACQUITY UPLC system (Waters, Milford, MA, USA) and a Synaptt Q-TOF (Waters, Manchester, UK) coupled with an electrospray ionization (ESI) source. The hydrolysate was obtained for enrichment, and confirmed by Nuclear Magnetic Resonance (NMR) (Bruker, Karlsruhe, Germany).

### 4.5. Determination of the Peak Areas of 3-O-EZ and Hydrolysate in EK and VEK

The powder of EK and VEK (about 2 g) were weighed and put into a stoppered conical flask. Then, 25 mL of ethyl acetate was carefully added, weighed and ultrasonicated (power 250 w, frequency 50 kHz) for 40 min. Furthermore, the loss of solvent was replenished with ethyl acetate, filtered and concentrated up to dryness. Lastly, the residue was dissolved in methanol and transferred to 2 mL volumetric flask. Hydrolysate was prepared individually at concentrations of 0.512 mg/mL. All solutions were stored at 4 °C until further analysis.

The mobile phase was composed of acetonitrile (A) and 0.1% formic acid in water (B) with a gradient elution as follows: 0–9 min, 40–59% A; 9–10 min, 59–64% A; 10–14 min, 64–80% A; 14–22 min, 80–99% A; 22–27 min, 99% A; 27–28 min, 99–40% A; 28–30 min, 40% A. The flow rate was set at 0.4 mL/min and the volume of injection was 2 μL.

### 4.6. Comparison of Toxicity of 3-O-EZ and Hydrolysate

#### 4.6.1. Acute Toxicity

The appropriate amount of 3-O-EZ and hydrolysate were weighed accurately and dissolved with 0.1% DMSO embryo medium (mixture of 15 mM NaCl, 0.5 mM KCl, 1 mM CaCl_2_, 1 mM MgSO_4_, 0.15 mM KH_2_PO_4_, 0.05 mM NH_2_PO_4_, and 0.7mM NaHCO_3_) [7]. Then, the concentrations of 3-O-EZ and hydrolysate solution were prepared at 1.41 and 1.16 mg/mL, respectively. The solution was stored at 4 °C until further analysis.

The six different concentration groups of 3-O-EZ (1.763, 1.058, 0.635, 0.381, 0.228, 0.137 μg/mL) and hydrolysate (116, 92.8, 74.24, 59.39, 47.51, 30.01 μg/mL) were set up. After 10 h, zebrafish embryos were seeded in a 24-well plate with ten zebrafish embryos per well. Zebrafish embryos were divided into three groups: Control group (0.1% DMSO embryo medium) (*n* = 20), 3-O-EZ group (*n* = 20) and hydrolysate group (*n* = 20). Then, the three groups were incubated in triplicate at 28 °C. The mortality date of zebrafish embryos at different drug concentrations was detected under a microscope after 1, 2, 3, and 4 days, and the LC_100_ and LC_0_ of 3-O-EZ and hydrolysate were obtained.

According to the LC_100_ and LC_0_ of 3-O-EZ, six concentration groups of 3-O-EZ (0.990, 0.693, 0.485, 0.339, 0.238, 0.166 μg/mL) were set up. After 10 h incubation, zebrafish embryos were seeded in 24-well plate with ten zebrafish embryos per well, and incubated in triplicate at 28 °C. The 3-O-EZ group was conducted in the same way as mentioned above. After 96 h incubation, the mortality of zebrafish embryos in each group was calculated.

#### 4.6.2. Developmental Toxicity

To further evaluate developmental toxicity (Heart rate, Pericardial edema and Scoliosis) of zebrafish embryos induced by 3-O-EZ and hydrolysate, high (0.412 μg/mL), moderate (0.206 μg/mL) and low (0.103 μg/mL) concentrations of 3-O-EZ and hydrolysate were given in triplicate for 96 h incubation at 28 °C. After 96 h incubation, zebrafish embryos were anesthetized and mounted on double concave slides in 3% methylcellulose. Then, the images were visualized under a DMi8 inverted microscope (Leica, Berlin, Germany), and the heart rate within 20 s were recorded.

#### 4.6.3. Cardiotoxicity

The treatment of 3-O-EZ and hydrolysate on zebrafish embryos was the same as “Developmental toxicity”, and SV-BA distance was measured using Image-Pro Plus 6.0 (IPP 6.0, Media Cybernetics, Silver Spring, MD, USA) software on the acquired images.

Meanwhile, zebrafish embryos were transferred into AO solution (4 mg/L) in darkness for 1.5 h at room temperature [21]. Then, the embryos were gently rinsed three times with phosphate buffered saline (PBS) (Gibco, Thermo Fisher, Waltham, MA, USA).

#### 4.6.4. Gastrointestinal Toxicity

The embryos were kept in the embryo medium for 96 h, and high (0.412 μg/mL), moderate (0.206 μg/mL) and low (0.103 μg/mL) concentrations of 3-O-EZ and hydrolysate were given in triplicate for 120 h incubation at 28 °C. After 120 h incubation, gastrointestinal peristalsis rates within 1 min were recorded under a microscope. And embryos were transferred into calcein solution (0.2%) in darkness for 10 min at room temperature [29]. Then, the embryos were gently rinsed three times with embryo medium, and gastrointestinal areas were measured using IPP 6.0 software on the acquired images.

#### 4.6.5. SDH and SOD Activity, GSH and MDA Content Analysis

The treatment of 3-O-EZ and hydrolysate on zebrafish embryos was the same as “Developmental toxicity”. After 96 h incubation, 50 embryos were randomly selected, and subsequently resuspended in 300 uL PBS. Then, the samples were centrifuged at 4000 rpm for 10 min at 4 °C (JX-FSTPRP-24, Jingxin, Shanghai, China) and the supernatant was recovered. The activities of SDH, SOD, and the content of MDA and GSH were measured using assay kits (Jiancheng, Nanjing, China). The levels of total protein were determined using a bicinchoninic acid (BCA) protein assay kit (Jiancheng, Nanjing, China).

#### 4.6.6. Caspase Activity as well as IL-2 and IL-8 Content Analysis

After 96 h incubation, 30 embryos were employed to measure the caspase-3 and caspase-9 activities. The enzyme activities were determined using assay kits (Jiancheng, Nanjing, China). In addition, the levels of IL-2 and IL-8 of embryos were quantified using enzyme-linked immunosorbent assay (ELISA) kits (Jiancheng, Nanjing, China).

#### 4.6.7. Gene Expression Analysis

Total RNA was extracted with Trizol from embryos. The transcription levels of genes related to oxidation, apoptosis, immunity and mitochondria were measured by quantitative real-time polymerase chain reaction (qRT-PCR). The primers are provided as Table 3. The 2^−ΔΔCt^ method was used to analyze the real-time PCR data.

### 4.7. Statistical Analysis

The LC_50_ was calculated by SPSS Statistics 17.0 software (International Business Machines Corporation, Armonk, NY, USA). One-way analysis of variance (ANOVA) followed by a Dunn’s multiple comparison test was used for comparing between control group and treated groups. It was considered to be statistically significant when the *p* value was below 0.05.

## 5. Conclusions

In summary, the results of the current study revealed that the content of 3-O-EZ was significantly reduced in VEK, while the levels of ingenol in VEK were significantly increased compared with EK. Moreover, acute toxicity, developmental toxicity and organic toxicity of 3-O-EZ and ingenol all demonstrated that ingenol exhibited less toxicity than 3-O-EZ. Together with our previous results on zebrafish embryos [7], this study contributes to our understanding of the mechanism of attenuation in toxicity of VEK, and provides the possibility of safe and rational use of EK in clinic.

## Figures and Tables

**Figure 1 molecules-24-03806-f001:**
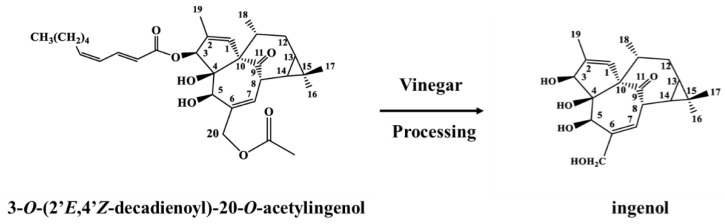
The conversion process between compounds 3-O-EZ and ingenol.

**Figure 2 molecules-24-03806-f002:**
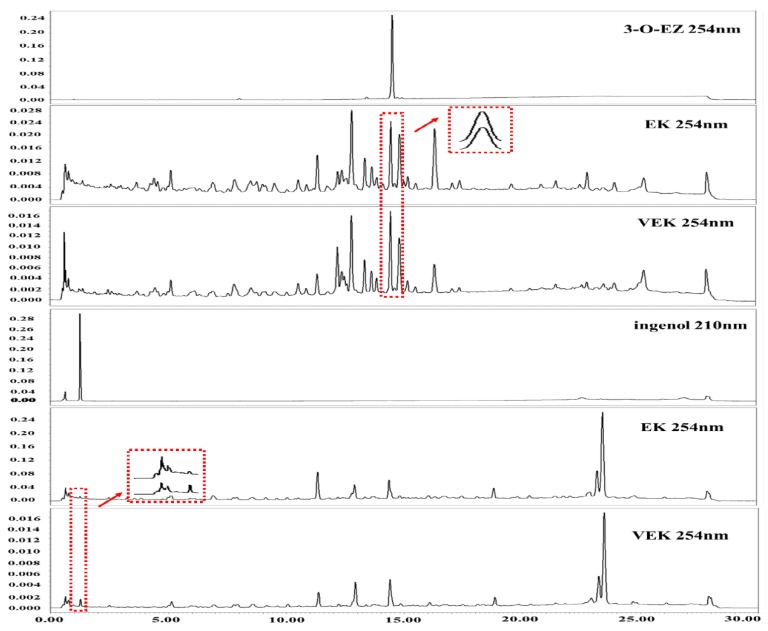
UPLC of the 3-O-EZ and ingenol in *Euphorbia kansui* (EK) and EK stir-fried with vinegar (VEK).

**Figure 3 molecules-24-03806-f003:**
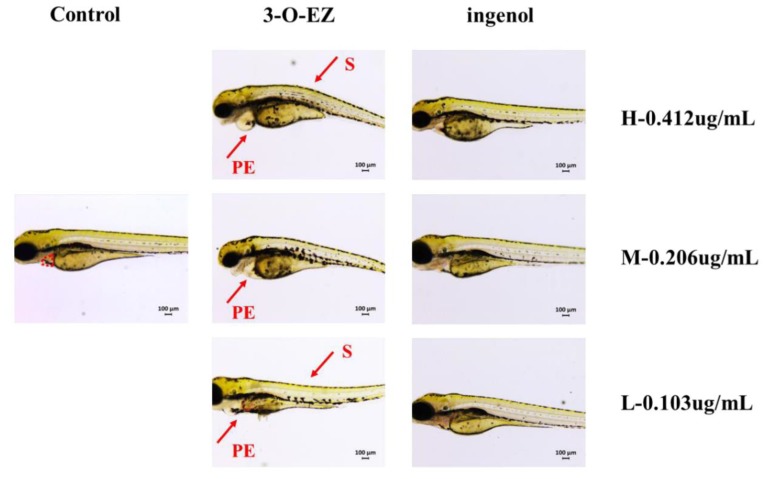
Abnormal embryos exposed to control, 3-O-EZ and ingenol groups. The control group was treated with the embryo medium, while the 3-O-EZ and ingenol groups were dissolved in the embryo medium with high (H, 0.412 ug/mL), moderate (M, 0.206 ug/mL) and low (L, 0.103 ug/mL) dose groups. Each treatment group showed varying degrees of pericardial edema (PE) and scoliosis (S).

**Figure 4 molecules-24-03806-f004:**
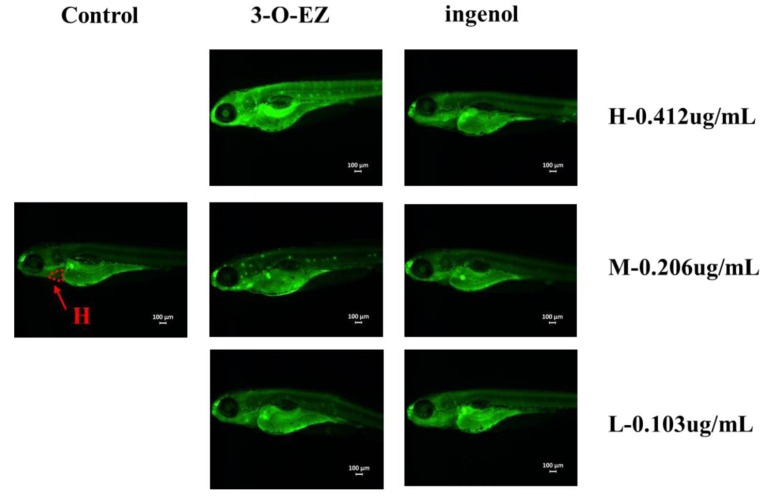
The results of AO staining in zebrafish embryos (heart, H).

**Figure 5 molecules-24-03806-f005:**
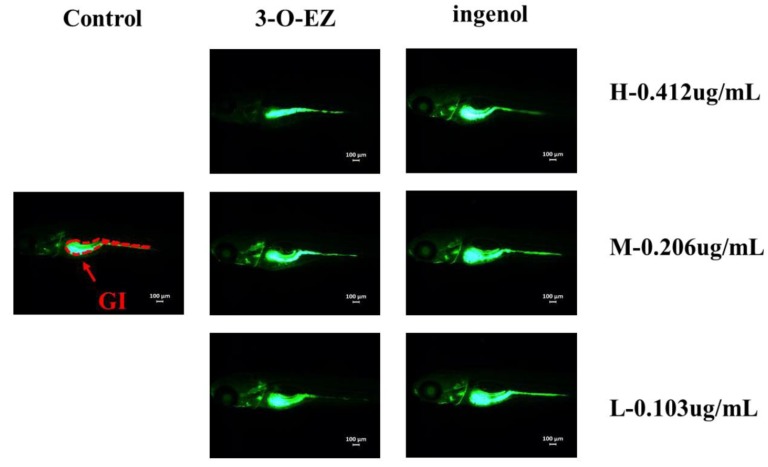
The results of calcein staining in the gastrointestinal (GI) tract of zebrafish embryos.

**Figure 6 molecules-24-03806-f006:**
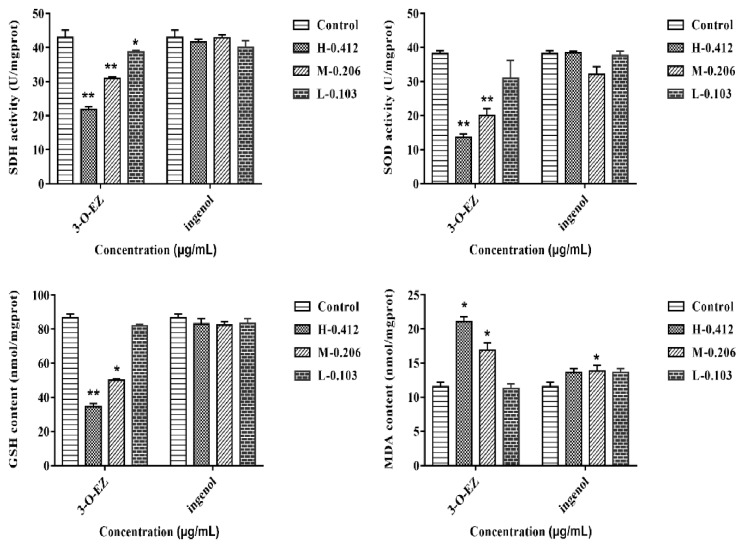
The enzyme activities of SDH and SOD, GSH and MDA content in zebrafish embryos (* *p* < 0.05, ** *p* < 0.01 compared to control group).

**Figure 7 molecules-24-03806-f007:**
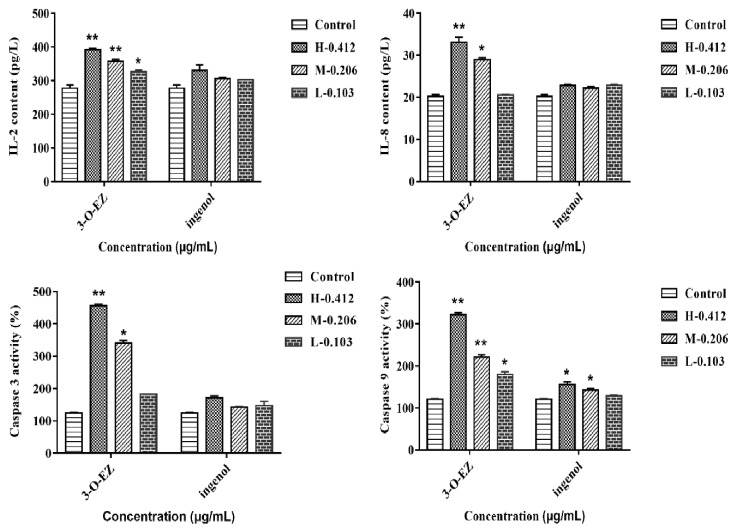
Caspase-3, Caspase-9 activities and IL-2, IL-8 content in zebrafish embryos. (* *p* < 0.05, ** *p* < 0.01 compared to control group).

**Figure 8 molecules-24-03806-f008:**
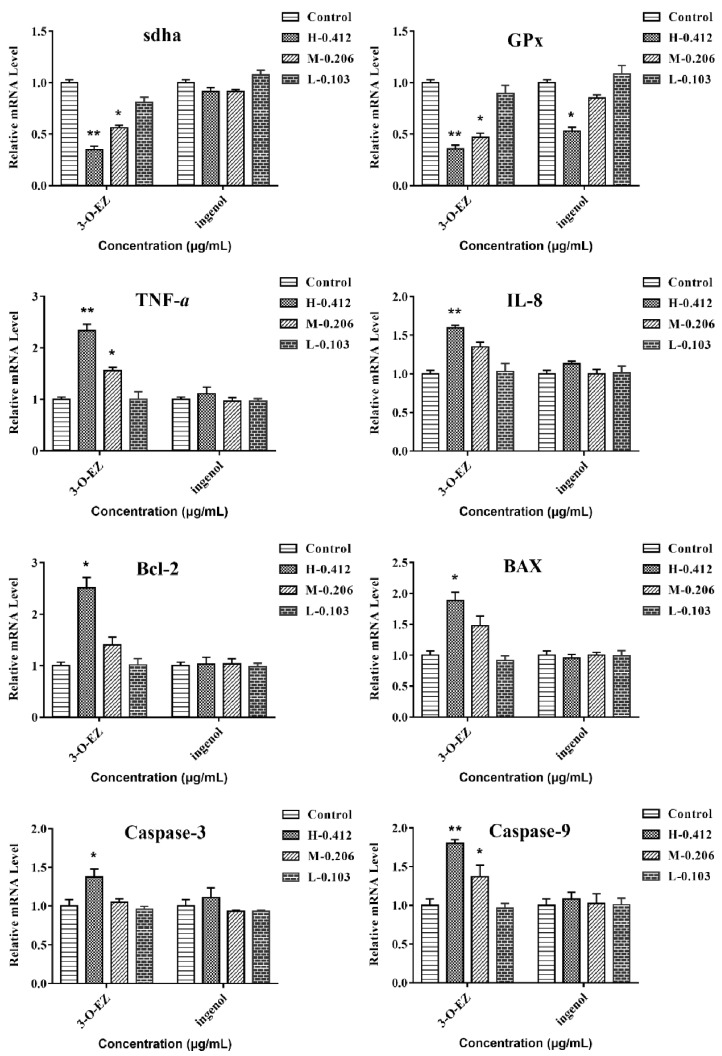
Expression of genes related to oxidative stress, immunity and apoptosis in zebrafish embryos (* *p* < 0.05, ** *p* < 0.01 compared to control group).

**Table 1 molecules-24-03806-t001:** The result of heart rate and SV-BA value in zebrafish embryos (mean ± SD, *n* = 10).

Group	Concentration (μg/mL)	Heart Rate	SV-BA
Control	0.1%(DMSO)	52.90 ± 1.52	171.09 ± 4.08
3-O-EZ	0.412	26.20 ± 1.99 **	286.32 ± 3.04 **
	0.206	38.80 ± 1.93 **	263.79 ± 2.23 **
	0.103	50.60 ± 1.83	193.24 ± 2.68 *
ingenol	0.412	51.60 ± 1.35	194.48 ± 1.48
	0.206	51.80 ± 1.99	171.62 ± 3.44
	0.103	52.50 ± 2.27	174.34 ± 2.38

* *p* < 0.05, ** *p* < 0.01 compared to control group.

**Table 2 molecules-24-03806-t002:** The result of gastrointestinal motility rate and area in zebrafish embryos (mean ± SD, *n* = 10).

Group	Concentration (μg/mL)	Gastrointestinal Motility Rate	Gastrointestinal Area
Control	0.1%(DMSO)	10.50 ± 0.97	97,771 ± 1073
3-O-EZ	0.412	3.90 ± 0.99 **	61,222 ± 2012 **
	0.206	7.20 ± 1.03 **	86,111 ± 1962 **
	0.103	9.20 ± 1.40	93,591 ± 1411 **
ingenol	0.412	10.10 ± 1.10	101,825 ± 1835
	0.206	9.80 ± 1.14	102,635 ± 2335
	0.103	10.60 ± 1.35	101,366 ± 1618

** *p* < 0.01 compared to control group.

**Table 3 molecules-24-03806-t003:** Sequences of the primer pairs used in the qRT-PCR.

Gene		Primer	References
β-actin	Forward	5′-AGAGCTATGAGCTGCCTGACG-3′	[30]
	Reverse	5′-CCGCAAGATTCCATACCCA-3′	
sdha	Forward	5′-TGGTATGCCGTTCAGCCGTA-3′	[23]
	Reverse	5′-GGCCAAGTCTTTGGCATTGG-3′	
GPx	Forward	5′-AGATGTCATTCCTGCACACG-3′	[23]
	Reverse	5′-AAGGAGAAGCTTCCTCAGCC-3′	
TNF-*α*	Forward	5′-GCTGGATCTTCAAAGTCGGGTGTA-3	[30]
	Reverse	5′-TGTGAGTCTCAGCACACTTCCATC-3′	
IL-8	Forward	5′-GTCGCTGCATTGAAACAGAA-3′	[23]
	Reverse	5′-CTTAACCCATGGAGCAGAGG-3′	
Bcl-2	Forward	5′-TCACTCGTTCAGACCCTCAT-3′	[30]
	Reverse	5′-ACGCTTTCCACGCACAT-3′	
Bax	Forward	5′-GGCTATTTCAACCAGGGTTCC-3′	[30]
	Reverse	5′-TGCGAATCACCAATGCTGT-3′	
Caspase-3	Forward	5′-CCGCTGCCCATCACTA-3′	[23]
	Reverse	5′-ATCCTTTCACGACCATCT-3′	
Caspase-9	Forward	5′-CTGAGGCAAGCCATAATCG-3′	[30]
	Reverse	5′-AGAGGACATGGGAATAGCGT-3′	

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
