# Peer review of "Toxicity Reduction of Euphorbia kansui Stir-Fried with Vinegar Based on Conversion of 3-O-(2′E,4′Z-Decadi-enoyl)-20-O-acetylingenol"

_molecules, 2019, doi:10.3390/molecules24203806_

Round 1
Reviewer 1 Report
In this manuscript, the authors describe the effect of treating Euphorbia nasui with vinegar in order to reduce the toxicity associated with its use as a TCM. Overall, I found the manuscript to be interesting and scientifically sound. However, there are a number of grammar and spelling mistakes as well as the incorrect word usage. This will require the help of a English speaking colleague or Professional editing service.
In general, the tables and figure legends do not contain enough information. They each need to contain enough information to stand alone. Additionally, the units should be defined for every measurement reported. Also, please ensure that the tables and figures are numbered correctly and that the text refers to the correct tables and figures. Currently, there are several places in the text where the authors refer to the wrong table. Lastly, please review your use of significant figures in reporting values in the tables. For example, in table 2 the values for gastrointestinal area should be reported at a minimum to whole numbers. With standard deviation of 1072, the values after the decimal point are totally meaningless.
Throughout the manuscript the authors lump together findings for SDH, SOD and GSH analysis, which they refer to as enzymatic activity. However, GSH is not an enzyme and thus has no activity. Please carefully review all reference to GSH and ensure that it is referred to as a concentration and not an activity.
Author Response
Dr. reviewer
Thank you for your comments concerning our manuscript entitled “Toxicity reduction of Euphorbia kansui stir-fried with vinegar based on conversion of 3-O-(2’E,4’Z-decadienoyl)-20-O-acetylingenol” at (Ref: molecules-617761) Molecules. Those comments are all valuable and very helpful for revising and improving our paper, as well as the important guiding significance to our researches. We have studied comments carefully and have made correction which we hope meet with approval. The responds to your comments are as flowing:
Comments of reviewer 1:
However, there are a number of grammar and spelling mistakes as well as the incorrect word usage. This will require the help of a English speaking colleague or Professional editing service.Reply: Thanks for the reviewer’s comment. We have made the improvements to the English language in our manuscript. The English types and grammar have been edited throughout by a native English speaker, please see manuscript-molecules-617761-Revised. Of course, we also hope to get help from professional English editors very much.
Currently, there are several places in the text where the authors refer to the wrong table. Lastly, please review your use of significant figures in reporting values in the tables. For example, in table 2 the values for gastrointestinal area should be reported at a minimum to whole numbers. With standard deviation of 1072, the values after the decimal point are totally meaningless.Reply: Thanks for the reviewer’s comment. The wrong tables had been revised, please see lines 116, 120, 128 and 136-143. And the values of gastrointestinal area in table 2 had been checked and revised, please see lines 143-144 (table 2).
Throughout the manuscript the authors lump together findings for SDH, SOD and GSH analysis, which they refer to as enzymatic activity. However, GSH is not an enzyme and thus has no activity. Please carefully review all reference to GSH and ensure that it is referred to as a concentration and not an activity.Reply: Thanks for the reviewer’s comment. The wrong expression of GSH activity had been corrected to GSH content, and the whole manuscript had been checked carefully, please see lines 34, 146-152,164, 213, 234, 331.
The revised part of the article has been marked with a red font. I hope you will find the improved manuscript worthy of publication in Molecules. Please let me know if any change is needed before it will be considered for publication.
Looking forward to hearing from you soon. Thanks for your time and consideration.
With kind personal regards,
Li Zhang, Professor
Nanjing University of Chinese Medicine
#138 Xianlin Avenue, Qi Xia District, Nanjing 210023, R. P. China
Tel: +86-13851472740
E-mail: zhangli@njucm.edu.cn
18 October 2019

Reviewer 2 Report
In the article entitled “Toxicity reduction of Euphorbia kansui (EK) stir-fried with vinegar based on conversion of 3-O-(2’E,4’Z-decadienoyl)-20-O-acetyl-ingenol” authors evaluated the 3-O-EZ and ingenol content in EK and in EK stir-fried with vinegar (VEK). Acute, developmental, cardiac and gastrointestinal toxicities were evaluated as well as the involvement of inflammatory response and apoptosis. Experiments have been well conducted and appropriately described. Results are clearly presented and the conclusions are supported by the results. However, I recommend an extensive review of English language which, in many parts of the manuscript, shows grammatical and syntactical errors.
Below, some suggestions to improve the manuscript:
Table 1 line 119 is incorrectly referred to as table 2. The formatting of tables does not allow a correct reading of the data reported in them. The addition of horizontal lines would greatly facilitate this.
Authors should show acute toxicity curves (see chapter 2.3.1) that allowed calculating the LC50 LC100 and LC0 values used in the further experiments.
Author Response
Dr. reviewer
Thank you for your comments concerning our manuscript entitled “Toxicity reduction of Euphorbia kansui stir-fried with vinegar based on conversion of 3-O-(2’E,4’Z-decadienoyl)-20-O-acetylingenol” at (Ref: molecules-617761) Molecules. Those comments are all valuable and very helpful for revising and improving our paper, as well as the important guiding significance to our researches. We have studied comments carefully and have made correction which we hope meet with approval. The responds to your comments are as flowing:
Comments of reviewer 2:
However, I recommend an extensive review of English language which, in many parts of the manuscript, shows grammatical and syntactical errors.Reply: Thanks for the reviewer’s comment. We have made the improvements to the English language in our manuscript. The English types and grammar have been edited throughout by a native English speaker, please see manuscript-molecules-617761-Revised. Of course, we also hope to get help from professional English editors very much.
Table 1 line 119 is incorrectly referred to as table 2. The formatting of tables does not allow a correct reading of the data reported in them. The addition of horizontal lines would greatly facilitate this.Reply: Thanks for the reviewer’s comment. Table 1 line 119 had been revised, please see lines 116-120, and the lines of manuscript had been added, please see manuscript-molecules-617761-Revised.
Authors should show acute toxicity curves (see chapter 2.3.1) that allowed calculating the LC50, LC100 and LC0 values used in the further experiments.Reply: Thanks for the reviewer’s comment. The acute toxicity curves of 3-O-EZ had been added in Supplementary Materials-Revised.
The acute toxicity curves of 3-O-EZ was shown in Figure S4, 3-O-EZ presented definite toxicity with LC100 values of 0.990 μg/ml and LC0 values of 0.166 μg/ml, and the LC50 values of 3-O-EZ was 0.412±0.01 μg/ml.
The revised part of the article has been marked with a red font. I hope you will find the improved manuscript worthy of publication in Molecules. Please let me know if any change is needed before it will be considered for publication.
Looking forward to hearing from you soon. Thanks for your time and consideration.
With kind personal regards,
Li Zhang, Professor
Nanjing University of Chinese Medicine
#138 Xianlin Avenue, Qi Xia District, Nanjing 210023, R. P. China
Tel: +86-13851472740
E-mail: zhangli@njucm.edu.cn
18 October 2019
